# Characterization in Inhibitory Effectiveness of Carbamazepine in Voltage-Gated Na^+^ and Erg-Mediated K^+^ Currents in a Mouse Neural Crest-Derived (Neuro-2a) Cell Line

**DOI:** 10.3390/ijms23147892

**Published:** 2022-07-17

**Authors:** Po-Ming Wu, Hsin-Yen Cho, Chi-Wu Chiang, Tzu-Hsien Chuang, Sheng-Nan Wu, Yi-Fang Tu

**Affiliations:** 1Department of Pediatrics, National Cheng Kung University Hospital, College of Medicine, National Cheng Kung University, Tainan 70101, Taiwan; bradsteve41@gmail.com; 2Institute of Clinical Medicine, College of Medicine, National Cheng Kung University, Tainan 70101, Taiwan; 3Department of Physiology, National Cheng Kung University Medical College, Tainan 70101, Taiwan; s36094083@gs.ncku.edu.tw (H.-Y.C.); s36091051@gs.ncku.edu.tw (T.-H.C.); 4Institute of Molecular Medicine, College of Medicine, National Cheng Kung University, Tainan 70101, Taiwan; chiangcw@mail.ncku.edu.tw; 5Institute of Basic Medical Sciences, National Cheng Kung University Medical College, Tainan 70101, Taiwan

**Keywords:** carbamazepine (Tegretol^®^, 5H-dibenzo[b,f]azepine-5-carboxamide), voltage-gated Na^+^ current, window Na^+^ current, persistent Na^+^ current, voltage-dependent hysteresis, *erg*-mediated K^+^ current, pulse train stimulation, neuroblastoma cell

## Abstract

Carbamazepine (CBZ, Tegretol^®^) is an anticonvulsant used in the treatment of epilepsy and neuropathic pain; however, several unwanted effects of this drug have been noticed. Therefore, the regulatory actions of CBZ on ionic currents in electrically excitable cells need to be reappraised, although its efficacy in suppressing voltage-gated Na^+^ current (*I*_Na_) has been disclosed. This study was undertaken to explore the modifications produced by CBZ on ionic currents (e.g., *I*_Na_ and *erg*-mediated K^+^ current [*I*_K(erg)_]) measured from Neuro-2a (N2a) cells. In these cells, we found that this drug differentially suppressed the peak (transient, *I*_Na(T)_) and sustained (late, *I*_Na(L)_) components of *I*_Na_ in a concentration-dependent manner with effective IC_50_ of 56 and 18 μM, respectively. The overall current–voltage relationship of *I*_Na(T)_ with or without the addition of CBZ remained unchanged; however, the strength (i.e., ∆area) in the window component of *I*_Na_ (*I*_Na(W)_) evoked by the short ascending ramp pulse (V_ramp_) was overly lessened in the CBZ presence. Tefluthrin (Tef), a synthetic pyrethroid, known to stimulate *I*_Na_, augmented the strength of the voltage-dependent hysteresis (Hys_(V)_) of persistent *I*_Na_ (*I*_Na(P)_) in response to the isosceles-triangular V_ramp_; moreover, further application of CBZ attenuated Tef-mediated accentuation of *I*_Na(P)_’s Hys_(V)_. With a two-step voltage protocol, the recovery of *I*_Na(T)_ inactivation seen in Neuro-2a cells became progressively slowed by adding CBZ; however, the cumulative inhibition of *I*_Na(T)_ evoked by pulse train stimulation was enhanced during exposure to this drug. Neuro-2a-cell exposure to CBZ (100 μM), the magnitude of *erg*-mediated K^+^ current measured throughout the entire voltage-clamp steps applied was mildly inhibited. The docking results regarding the interaction of CBZ and voltage-gate Na^+^ (Na_V_) channel predicted the ability of CBZ to bind to some amino-acid residues in Na_V_ due to the existence of a hydrogen bond or hydrophobic contact. It is conceivable from the current investigations that the *I*_Na_ (*I*_Na(T)_, *I*_Na(L)_, *I*_Na(W)_, and *I*_Na(P)_) residing in Neuro-2a cells are susceptible to being suppressed by CBZ, and that its block on *I*_Na(L)_ is larger than that on *I*_Na(T)_. Collectively, the magnitude and gating of Na_V_ channels produced by the CBZ presence might have an impact on its anticonvulsant and analgesic effects occurring in vivo.

## 1. Introduction

Carbamazepine (CBZ, Tegretol^®^, 5H-dibenzo[b,f]azepine-5-carboxamide) is an aromatic anticonvulsant that has been widely used for the treatment of seizure disorders and neuropathic pain specifically for trigeminal neuralgia [1,2,3,4,5]. This drug has been also demonstrated as an adjunctive treatment in schizophrenia or myotonia and as a second-line agent in bipolar disorder [6,7,8,9,10].

Of additional notice, although CBZ is safe and effective in anti-convulsant activities, the unwanted events following CBZ treatment, such as hyponatremia, QT-interval prolongation, hyperprolactinemia, change in pitch perception and idiosyncratic reactions, have gradually emerged [5,11,12,13,14,15,16,17,18,19,20,21,22,23,24,25,26,27,28]. The effectiveness of CBZ in inhibiting the activity of ATP-sensitive K^+^ (K_ATP_) channels has also been demonstrated [29,30]. However, this drug was reported to restore neuronal signaling, protein synthesis, and cognitive function in a mouse model of fragile X syndrome [31] as well as to improve motor impairment either in myotonia congenita or in the model of Machado-Joseph disease [9,10]. Therefore, it is valuable to reappraise the ionic mechanism of CBZ actions in different types of transmembrane ionic currents, particularly the voltage-gated Na^+^ (Na_V_) channels, which were recently explored for their therapeutic or pharmacological effectiveness [1,4,32,33,34,35].

It has been established that nine isoforms (i.e., Na_V_1.1-1.9 [or SCN1A-SCN5A and SCN8A-SCN11A]) of Na_V_ channels are widely distributed in mammalian excitable tissues located in the central or peripheral nervous system and neuroendocrine system [36]. The activity of these channels is to depolarize the cell and to generate the upstroke of the action potential, thereby controlling the firing amplitude, frequency, and pattern inherently in electrically excitable cells [36,37,38]. Of additional note, some inhibitors of Na_V_ channels (e.g., esaxerenone, ranolazine, and sparsentan) were recognized to increase the inactivation rate of voltage-gated Na^+^ current (*I*_Na_) [37,38,39,40,41,42], whereas several activators of Na_V_ channels (e.g., tefluthrin [Tef]) could preferentially slow the inactivation rate as well as increase the late component of *I*_Na_ (*I*_Na(L)_) [43,44,45].

Therefore, in the current study, the electrophysiological effects of CBZ and other related compounds in Neuro-2a cells were investigated. Neuro-2a cells were chosen because they mainly express Na_V_1.2, 1.3, 1.4, and 1.7 [46]. Na_V_1.2 and 1.3 are primarily expressed in the central nervous system and involved in the pathomechanisms of epilepsy, while Na_V_1.7 is in the peripheral nervous system and plays a role in neuropathic pain [36,37,38]. We sought to (1) determine if CBZ has any effects on the peak (transient, *I*_Na(T)_) and sustained (late, *I*_Na(L)_) components of *I*_Na_ residing in these cells; (2) examine if this drug affects the windows component of *I*_Na_ (*I*_Na(W)_); (3) study if the voltage-dependent hysteresis (Hys_(V)_) of the persistent *I*_Na_ (*I*_Na(P)_) can be altered by its presence; (4) explore if cell exposure to CBZ can result in the recovery of *I*_Na(T)_ inactivation as well as the cumulative inhibition of *I*_Na(T)_; and (5) determine its effects on erg-mediated K^+^ current (*I*_K(erg)_). The present results revealed that the differential inhibition of *I*_Na(T)_ and *I*_Na(L)_ by CBZ as well as its actions on the magnitude, gating and Hys_(V)_ behavior of *I*_Na_ may converge to engage in its regulation of the electrical behaviors of excitable cells (e.g., Neuro-2a cells).

## 2. Results

### 2.1. Modification by the CBZ Presence of Voltage-Gated Na^+^ Current (I_Na_) Measured in Neuro-2a Cells

In the initial stage of the experiments, we exploited the whole-cell configuration of the patch-clamp technique to explore it the addition of CBZ could lead to any perturbations on ionic currents (e.g., *I*_Na_) residing in Neuro-2a cells. Extracellular Ca^2+^ was recently reported to alter the voltage dependence of I_Na_ [47]. In order to measure the current flowing through *I*_Na_, we put the cells into a Ca^2+^-free Tyrode’s solution which contained 0.5 mM CdCl_2_ and 10 mM tetraethylammonium chloride (TEA), while the recording pipette was then backfilled with a Cs^+^-enriched internal solution. As the whole-cell mode was firmly established, in order to evoke INa in Neuro-2a cells, we held the tested cell at the potential of −80 mV, and a hyperpolarizing pulse of −100 mV for a duration of 30 ms was then applied to precede the depolarizing command voltage from −100 to −10 mV for another 30 ms. Under these experimental conditions, we were able to detect the emergence of a fast inward current (i.e., inward flux of cations) which exhibited the rapidly activating and inactivating time course (Figure 1). This type of transient inward current stimulated by such a brief rectangular pulse was owing to the activation of a TTX-sensitive voltage-gated Na^+^ current (I_Na_), which is sensitive to being inhibited or stimulated by tetrodotoxin (1 μM) or tefluthrin (Tef, 10 μM), respectively. With Neuro-2a cell exposure to 1 μM, the amplitude of *I*_Na_ measured at the beginning of the short depolarizing pulse decreased from 813 ± 23 to 52 ± 4 pA (n = 8, t value = 3.7, *p* < 0.01). It was hence identified as a TTX-sensitive voltage-gated Na^+^ current (I_Na_) [38,40,42,43,44,45,48,49].

As demonstrated in Figure 1A, one minute after Neuro-2a cells were exposed to CBZ, the amplitude of peak *I*_Na_ (or transient *I*_Na_, [*I*_Na(T)_]) progressively decreased along with a concurrent reduction in the inactivation time constant of the current. The addition of CBZ at a concentration of 30 or 100 μM decreased *I*_Na(T)_ amplitude to 702 ± 23 pA (n = 8; t value = 3.1; *p* < 0.05) or 414 ± 19 pA (n = 8; t value = 2.9; *p* < 0.05), respectively, from a control value of 839 ± 33 pA (n = 8). Additionally, in the presence of 100 μM CBZ, the time constant in the slow component of the current inactivation (τ_inact(S)_) was concurrently shortened to 1.6 ± 0.2 ms (n = 8; t value = 3.0; *p* < 0.05) from a control value of 2.3 ± 0.3 ms (n = 8), whereas no clear difference in the fast component of I_Na(T)_ inactivation was demonstrated. After CBZ was removed, the *I*_Na(T)_ amplitude was returned to 831 ± 31 pA (n = 8). Furthermore, upon exposure to ranolazine (10 μM), the peak amplitude of I_Na(T)_ was decreased from 823 ± 24 to 324 ± 17 pA (n = 7; t value = 3.1; *p* < 0.05), which was consistent with the finding that ranolazine was an inhibitor of the late component of I_Na_ (I_Na(L)_) [39,40].

We further constructed the relationship between the CBZ concentration and the *I*_Na(T)_ or I_Na(L)_ evoked in response to an abrupt depolarizing pulse. In this stage of the measurements, each cell was rapidly stepped from −100 to −10 mV, and the amplitude of *I*_Na(T)_ or *I*_Na(L)_ acquired at different CBZ concentrations (3 μM–1 mM) was collated; the results are presented in Figure 1B. It is clear from the present observations that the addition of CBZ results in a concentration-dependent reduction in *I*_Na(T)_ and *I*_Na(L)_ amplitudes in Neuro-2a cells. According to a modified Hill equation, the IC_50_ values needed for CBZ-mediated inhibition of *I*_Na(T)_ and *I*_Na(L)_ seen in Neuro-2a cells were calculated to be 56 and 18 μM, respectively. The results, therefore, reflect that the CBZ presence is capable of exerting a depressant action on the depolarization-activated *I*_Na_ that is concentration-dependently observed in Neuro-2a cells. Of additional note, the amplitude of *I*_Na(L)_ was subject to being inhibited to a greater extent than that of *I*_Na(T)_ during the exposure to CBZ.

### 2.2. Effect of CBZ on the Quasi-Steady-State Current–Voltage (I–V) Relationship of I_Na(T)_

We further examined the steady-state *I–V* relationship of *I*_Na(T)_ with or without CBZ treatment. These voltage clamp experiments were conducted in cells held at −80 mV, and a series of command voltages between −80 and +40 mV were applied to the tested cells. As demonstrated in Figure 2, CBZ (30 μM) did not alter the overall *I–V* relationship of *I*_Na(T)_, although the *I*_Na(T)_ amplitude can be decreased in its presence. The *I–V* curves acquired in the control period and during cell exposure to 10 μM CBZ were approximately fitted with Boltzmann function. In control (i.e., CBZ was not present), G = 36.1 ± 1.2 nS, V_h_ = −24.1 ± 1.1 mV, k = 14.0 ± 0.9 (n = 8), while in the presence of 10 μM CBZ, G = 20.8 ± 1.1 nS, V_h_ = −24.4 ± 1.1 mV, k = 14.3 ± 0.9 (n = 8; t value =1.61; *p* > 0.05). It is, therefore, reasonable to assume that the overall quasi-steady-state activation curve of *I*_Na(T)_ remained unaltered during cell exposure to 10 μM CBZ, although the *I*_Na(T)_ magnitude was decreased in its presence.

### 2.3. Effect of CBZ on the Window Component of I_Na_ (I_Na(W)_) Recorded from Neuro-2a Cells

The presence of instantaneous *I*_Na(W)_ evoked in response to the upsloping (or ascending) ramp voltage (V_ramp_) was demonstrated earlier in different types of electrically excitable cells [48,50]. We next continued to explore whether the CBZ existence could modify the magnitude of *I*_Na(W)_ evoked by rapid ascending V_ramp_. For performing these experiments, we voltage-clamped the tested cell at −80 mV and applied an ascending V_ramp_ from −100 to +10 mV for 50 ms (with a ramp speed of 2.2 mV/ms) to evoke *I*_Na(W)_. Within one minute of exposing cells to CBZ (10 or 30 μM), the strength (∆area) of *I*_Na(W)_ acquired by the 50 ms upsloping V_ramp_ was strikingly decreased (Figure 3). Treating the cell with 10 or 30 μM CBZ resulted in a measurable reduction of the *I*_Na(W)_’s ∆area to 4.51 ± 0.65 or 2.33 ± 0.43 mV·nA (n = 8, t value = 3.1, *p* < 0.05), respectively, from a control of 8.35 ± 0.97 mV·nA (n = 8). In the continued presence of 30 μM CBZ, subsequent addition of tefluthrin (10 μM, Tef) attenuated CBZ-mediated decrease of ∆area, as demonstrated by an increase of ∆area value to 5.41 ± 0.86 mV·nA (n = 8; t value = 3.2; *p* < 0.05). Tef has been shown to be an activator of *I*_Na_ [43,44,45,48].

### 2.4. Effect of CBZ on the Hysteretic Behavior of Persistent Na^+^ Current (I_Na(P)_) Triggered by Upright Isosceles-Triangular V_ramp_

The background Na^+^ currents have been growingly demonstrated in different types of excitable cells [37,43,51,52,53,54,55]. Earlier investigations have also shown the effectiveness of voltage-dependent hysteresis (Hys_(V)_) of *I*_Na(P)_ in modifying electrical behaviors in many types of excitable cells [48,49]. Therefore, we wanted to determine whether and how CBZ could modify the strength of *I*_Na(P)_’s Hys_(V)_ activated by an upright isosceles-triangular V_ramp_. We voltage-clamped the tested cell at −80 mV and applied an upsloping (ascending) limb from −100 to +50 mV to the cell followed by a downsloping (descending) limb back to −100 mV (i.e., upright isosceles-triangular V_ramp_ and a ramp speed of ±0.5 mV/ms) for a total duration of 600 ms. Under our experimental conditions, the Hys_(V)_ behavior of *I*_Na(P)_ in response to such double V_ramp_ was observed as a striking figure-of-eight (i.e., ∞-shaped configuration in the instantaneous *I–V* relationship of *I*_Na(P)_ acquired in the presence of 10 μM tefluthrin (Tef) or 10 μM Tef plus 30 μM CBZ (Figure 4A). In other words, during cell exposure to 10 μM Tef, two distinct loops were noted; that is, the *I*_Na(P)_ at a high- (i.e., in counterclockwise direction) threshold loop and a low- (i.e., clockwise direction) threshold loop, activated by the upsloping and downsloping limbs of the upright double V_ramp_. Notably, as shown in Figure 4B,C, in the presence of 10 μM Tef alone, the amplitudes of *I*_Na(P)_ responding to both rising limb of double V_ramp_ at the level of −20 mV and falling limbs at −60 mV were 661 ± 52 and 162 ± 29 pA (n = 8), respectively. As cells were continually exposed to 10 μM Tef, further addition of 30 μM CBZ decreased current amplitudes at the same level of membrane potential to 511 ± 46 and 91 ± 23 pA (n = 8; *p* < 0.05). As such, findings from the present data enabled us to show an emergence of Hys_(V)_ behavior for *I*_Na(P)_ activation in response to double V_ramp_, and that the Hys_(V)_ strength of the current was raised by adding Tef. Moreover, under the exposure to Tef, the Hys_(V)_ strength of the *I*_Na(P)_ observed in Neuro-2a cells was subject to being attenuated by further application of CBZ (10 or 30 μM).

### 2.5. Effect of CBZ on the Recovery from I_Na(T)_ Inactivation Evoked during Varying Interpulse Intervals

Next, a two-step voltage protocol in which the interpulse interval increases with a geometrics-based progression (common ratio = 2) was applied to the tested cell to see whether CBZ leads to any adjustments on the recovery of *I*_Na(T)_ from inactivation. In this protocol, a 30 ms step from −80 to −10 mV (prepulse, the first pulse) was first applied and followed by another 30 ms step to −10 mV (test pulse, the second pulse) to inactivate most of the current during various interpulse interval (i.e., interval between the first and second pulse). The recovery from current inactivation was then examined at different time points with a geometrics-based progression at the holding potential of −80 mV, as presented semi-logarithmically in Figure 5. The time constants of recovery from *I*_Na(T)_ inactivation acquired in the absence and presence of 10 μM CBZ were least-squares fitted by a single-exponential with the values of 75.3 ± 2.3 and 223.1 ± 6.2 ms (n = 8; *p* < 0.05), respectively. The experimental observations thus indicate that there was a measurable prolongation in the recovery from *I*_Na(T)_ inactivation as Neuro-2a cells were exposed to CBZ.

### 2.6. CBZ-Induced Increase in Cumulative Inhibition of I_Na(T)_ Inactivation in Neuro-2a Cells

The *I*_Na(T)_ inactivation could be accumulated during repetitive short pulses [48,56,57,58]. Thus, additional measurements were taken to see whether CBZ could modify the inactivation process of *I*_Na(T)_ evoked in a train of depolarizing stimuli. The tested cell was held at −80 mV, and the stimulus protocol, i.e., repetitive depolarization to −10 mV (20 ms in each pulse with a rate of 40 Hz for 1 s), was applied to it. The *I*_Na(T)_ inactivation seen in Neuro-2a cells was evoked by a 1 s repetitive depolarization from −80 to −10 mV with a decaying time constant of 58.4 ± 3.8 ms (n = 7) in the control period (i.e., CBZ was not present) (Figure 6). That is, there is a rapid current decay in a single-exponential fashion. Of additional note, during exposure to 30 μM CBZ, the exponential time course of *I*_Na(T)_ evoked by the same train of depolarizing pulses was conceivably reduced to 21.1 ± 3.2 ms (n = 7; t value = 3.1; *p* < 0.05), in addition to a decrease in *I*_Na(T)_ amplitude. Under this scenario, the experimental observations indicate that, apart from its reduction in current magnitude, under cell exposure to CBZ, the decaying time course of *I*_Na(T)_ elicited by a 1 s train of a depolarizing pulse (i.e., accumulative inactivation of the current) can be overly enhanced in these cells.

### 2.7. Inhibition of Erg-Mediated K^+^ Current (I_K(erg)_) Caused by CBZ

Some of small-molecule *I*_Na_ inhibitors may influence the magnitude of *I*_K(erg)_ [39]. Therefore, in another separate set of experiments, we evaluated whether the CBZ presence could alter *I*_K(erg)_ identified in Neuro-2a cells. In order to measure *I*_K(erg)_, we placed the cells to be immersed in a high-K^+^, Ca^2+^-free external solution which contained 1 μM TTX and 0.5 mM CdCl_2_, and we filled up the recording pipette by using a K^+^-enriched (145 mM) internal solution. In these experiments, we held the examined cell at −10 mV and a series of command voltages ranging between −100 and 0 mV was thereafter applied to it for a duration of 1 s. As shown in Figure 7A, under this voltage clamp protocol, a family of *I*_K(erg)_ was robustly elicited, and the currents were manifested by an inwardly directed rectifying property with a large relaxation in the time course of current deactivation as described previously [59,60,61]. Figure 7B illustrates the average *I–V* relationship of the peak (square symbols) and sustained (circle symbols) components for deactivating *I*_K(erg)_ acquired with or without cell exposure to 100 μM CBZ. For example, in the control period (i.e., CBZ was not present), the peak and sustained components of *I*_K(erg)_ at the level of −100 mV were 780 ± 53 and 112 ± 34 pA, respectively (n = 7), while cell exposure to 100 μM CBZ significantly decreased peak and sustained *I*_K(erg)_ to 568 ± 48 and 58 ± 20 pA, respectively (n = 7; t value = 3.2; *p* < 0.05). Moreover, as cells were continually exposed to 100 μM CBZ, a further application of 30 μM diazoxide [30,62], an activator of ATP-sensitive K^+^ channels, was unable to reverse CBZ-induced inhibition of *I*_K(erg)_. Therefore, the addition of CBZ could result in a mild inhibition of *I*_K(erg)_ in response to long-lasting membrane hyperpolarization.

## 3. Discussion

The noticeable findings in this study provide evidence that the presence of CBZ, known to exert anticonvulsant and analgesic activities, was able to exert a depressant action on *I*_Na(T)_ and *I*_Na(L)_ in Neuro-2a cells in a concentration-dependent manner. The *I*_Na_ (*I*_Na(L)_) in response to short step depolarization is suppressed to a greater extent than the *I*_Na(T)_; therefore, the estimated IC_50_ values required for the inhibition of *I*_Na(T)_ and *I*_Na(L)_ seen in these cells were 56 and 18 μM, respectively. The overall *I-V* relationship of *I*_Na(T)_ remained unchanged during Neuro-2a-cell exposure to CBZ. The magnitude of *I*_Na(W)_ in response to short ascending V_ramp_ was diminished by adding CBZ. The Hys_(V)_ strength of *I*_Na(P)_ activated by upright isosceles-triangular V_ramp_ was overly augmented by cell exposure to Tef, and, in continued presence of Tef, further addition of CBZ overcame Tef-stimulated Hys_(V)_ strength of *I*_Na(P)_. The recovery of *I*_Na(T)_ inactivation emerging during varying interpulse intervals became slowed in the CBZ presence; however, the cumulative inhibition of *I*_Na(T)_ evoked by pulse train stimulation was enhanced by adding this drug. The CBZ presence also caused a mild inhibitory effect on *I*_K(erg)_ amplitude. Taken together, the activity of Na_V_ channels in excitable cells (e.g., Neuro-2a cells) may conceivably confer the susceptibility to modifications by CBZ. The differential inhibition by CBZ of *I*_Na(T)_ and *I*_Na(L)_ is of particular significance and may participate in its regulation of the electrical behaviors of excitable cells occurring in vivo [4,7,8,9,10,31,35].

In keeping with previous observations [35,63,64,65,66], the *I*_Na_, which displayed with the rapidly activating and inactivating time course (Figure 1), was not only functionally active in Neuro-2a cells, but also susceptible to being inhibited or stimulated by the presence of CBZ or Tef, respectively. Furthermore, we extended to provide evidence showing that the *I*_Na(T)_ and *I*_Na(L)_ residing in Neuro-2a cells were differentially inhibited by cell exposure to this drug. Our study is thus in accordance with earlier studies [33], showing that the resurgent *I*_Na_ is sensitive to being inhibited by the CBZ presence.

In the current study, we observed the non-linear Hys_(V)_ of *I*_Na(P)_ in the presence of Tef (10 μM) or Tef (10 μM) plus CBZ (10 or 30 μM), by use of the upright isosceles-triangular V_ramp_ (Figure 4). The Tef presence was noticed to accentuate Hys_(V)_ strength of *I*_Na(P)_ as stated previously [38,49]. Also of note, under cell exposure to Tef plus CBZ, such Hys_(V)_ behavior became attenuated. That is, the peak *I*_Na(P)_ activated by the ascending (upsloping) limb of the triangular V_ramp_ was overly decreased, particularly at the level of −20 mV, while the *I*_Na(P)_ amplitude at the descending (downsloping) end (e.g., at −60 mV) was concurrently reduced. Moreover, the instantaneous figure-of-eight (i.e., infinity-shaped; ∞) residing in the Hys_(V)_ loop that is activated by triangular V_ramp_ was observed, indicating that there is a counterclockwise direction in the high-threshold loop (i.e., a relationship of *I*_Na(P)_ amplitude as a function of membrane potential), followed by a clockwise direction in the low-threshold loop during activation [41,48,49,67]. Therefore, there were two types of Hys_(V)_’s loop. One high-threshold loop with a peak at −20 mV and one low-threshold loop with a peak at around −60 mV. In the presence of Tef, CBZ was still able to attenuate the Hys_(V)_ strength of *I*_Na(P)_. Findings from these observations, therefore, reveal that the double V_ramp_-induced *I*_Na(P)_ undergoes striking Hys_(V)_ change in the voltage dependence, and that such Hys_(V)_ loops are subjected to attenuation by adding CBZ. It also needs to be emphasized that the Hys_(V)_ behavior of *I*_Na(P)_ demonstrated here could be strongly linked to the magnitude of background Na^+^ currents occurring in different types of excitable cells [43,51,52,53,54,55]. Additional research needs to be conducted to understand whether CBZ-mediated changes in Hys_(V)_ behavior are tightly linked to conformational changes in the voltage sensors of the Na_V_ channel [68].

The time-dependent decline of *I*_Na(T)_ during a 40 Hz train of depolarizing voltage steps (i.e., 20 ms pulses applied from −80 to −10 mV at a rate of 40 Hz for a total duration of 1 s) was observed in this study, indicating that there is use dependence of *I*_Na(T)_ during repetitive depolarization as demonstrated recently [48,57,58]. Moreover, such exponential decrease in *I*_Na(T)_ responding to pulse train stimulation became overly pronounced in the presence of CBZ. It is, therefore, plausible to assume that cell exposure to CBZ would result in a loss-of-function change caused by the increased time course of *I*_Na(T)_ inactivation, and that CBZ-mediated reduction of *I*_Na(T)_ is linked to substantial use-dependent decrease in *I*_Na(T)_ during rapid repetitive stimuli or high frequency firing [56,57].

In the current observations, we revealed that Neuro-2a cell exposure to CBZ could decrease the recovery of *I*_Na(T)_ inactivation evoked by various interpulse intervals in a geometrics-based progression, although it increased the inactivation current evoked during pulse-train stimulation. Therefore, the post-spike *I*_Na(T)_ during rapid repetitive stimuli is closely linked to the increased inactivation in Neuro-2a cells, indicating that such fast inactivation develops from open channels during exposure to CBZ. Moreover, if recovery from current inactivation occurs through the open state (conformation), it would be accompanied by a decrease in small residual steady Na^+^ current (e.g., *I*_Na(P)_) in the CBZ presence. The CBZ presence would therefore be anticipated to decrease the magnitude of post-spike and steady currents, hence diminishing the occurrence of subthreshold potential [56]. Likewise, the magnitude of V_ramp_-induced *I*_Na(W)_ seen in Neuro-2a cells was expected to become lessened during exposure to CBZ. It is, therefore, plausible to assume that the CBZ molecule may have a higher affinity to the open/inactivated state than to the closed (resting) state residing in the Na_V_ channels, although the detailed ionic mechanisms of its inhibitory actions on the channel warrant further investigations.

The observations presented herein are somewhat different from previous reports, showing that the *I*_Na(L)_ was little affected by adding CBZ, despite its ability to inhibit *I*_Na(T)_ [32,34,35]. However, consistent with previous reports [65,66], the *I*_Na_ magnitude which can be functionally expressed in Neuro-2a cells was additionally noticed to be susceptible to being differentially inhibited by CBZ. One of the reasons for this discrepancy could be different isoforms of the Na_V_ channel α-subunit residing in different cell types. For example, Neuro-2a cells expressed Na_V_1.2, 1.3, 1.4, and 1.7 [64], while the murine osteoblasts used by Petty et al. [35] expressed Na_V_1.2, 1.3, and 2.1. Whatever the ionic mechanism involved, it is likely that the CBZ molecule may have the propensity to exert a higher affinity to the open/inactivated state than to the resting (closed) state residing in the Na_V_ channel, thereby leading to a destabilization of open conformation.

The well-known function of Erg-mediated potassium currents *I*_K(erg)_ is its contribution to the repolarization of the heart action potential [59]. This can also be recorded in various excitable cells, such as neuroendocrine cells and neuroblastoma cells (ex. Neuro-2a cells). In these excitable cells, *I*_K(erg)_ serves the function of a threshold current, which modulates cell excitability indirectly relevant to seizure susceptibility [59]. In this study, we found that CBZ could slightly inhibit *I*_K(erg)_ in response to long-lasting membrane hyperpolarization in Neuro-2a cells. Earlier investigations have demonstrated the capability of CBZ to decrease the activity of ATP-sensitive K^+^ channels [29,30]. However, during whole-cell configuration made under our experimental conditions, the ATP concentration of the pipette internal solution was 3 mM, a value that adequately blocks most K_ATP_-channel activity. Moreover, in continued presence of CBZ, further addition of diazoxide failed to attenuate CBZ-mediated inhibition of *I*_K(erg)_. Diazoxide is an opener of K_ATP_ channels [30,62]. It, therefore, seems unlikely that the inhibitory effect of CBZ on *I*_K(erg)_ magnitude seen in Neuro-2a cells is mainly due to its activity as a regulator of K_ATP_ channels.

Differential concentration-dependent inhibition of *I*_Na(T)_ and *I*_Na(L)_ with effective IC_50_ values of 56 and 18 μM was, respectively, found during the exposure to CBZ. As Neuro-2a cells were exposed to CBZ, the amplitude of *I*_Na(T)_ was also decreased in combination with a substantial decrease in the τ_inact(S)_ value of the current evoked in response to the short depolarizing pulse. The previous pharmacokinetic studies have shown that following one hour of administration with CBZ, its plasma level could reach concentrations ranging between 2 and 14 μg/mL (or 8.5 and 59 μM) [69,70]. The observed and predicted CBZ concentrations were also found within therapeutic windows (4–12 μg/mL or 17–51 μM) [70]. Moreover, the actions of CBZ on membrane excitability could be heavily dependent on various factors, including the CBZ concentration used, the pre-existing level of resting potential, different firing patterns of action potentials, and a combination of these three. Therefore, the CBZ actions on *I*_Na_ demonstrated herein are most likely to be therapeutically or pharmacologically relevant. However, the presence of adenosine (30 μM) alone did not cause any effects on the magnitude of *I*_Na_ seen in Neuro-2a cells, reflecting that CBZ-induced inhibition of *I*_Na_ could be direct and independent of its binding to adenosine receptors as stated previously [71].

CBZ has been increasingly demonstrated to cause hyponatremia, which mimics the syndrome of inappropriate antidiuretic hormone secretion [11,12,13,14,15,16,18,19,20,21,22,23,24,27]. Esaxerenone, known to be a nonsteroidal mineralocorticoid receptor blocker, was demonstrated to modify the magnitude and gating of *I*_Na_ [38], while sparsentan, a dual antagonist of endothelial type A receptor and angiotensin II receptor, could suppress *I*_Na_ [41]. It is interesting to note that the extent to which CBZ-mediated inhibition of *I*_Na(T)_ and *I*_Na(L)_ observed in Neuro-2a cells can contribute to its consequent effect on hyponatremia in a state in which lowered intracellular Na^+^ concentrations occur remains to be delineated. However, how the impact of CBZ on *I*_Na(T)_, *I*_Na(L)_, *I*_Na(P)_, or *I*_Na(W)_ shown herein might influence other adverse effects of this compound is largely unknown, and it needs to be further investigated.

Here, we additionally investigated how the protein of hNa_V_1.7 could be docked with CBZ by using PyRx software. The predicted binding sites of the CBZ were demonstrated in Figure 8. Notably, as it is docked to hNa_V_1.7, CBZ can form a hydrogen bond with residue Asp 1722 with a distance of 3.10 Å. The detailed structure of this hNa_V_ channel, which is a good exemplar for hNa_V_ pharmacology, was shown in a recent study [72]. Moreover, CBZ can form hydrophobic contact with several residues, including Ile 1622, Leu 1626, Asn 1714, Val 1717, Ala 1718, Val 1721, and Ala 1723. These results tempted us to reflect that CBZ can directly bind to the amino acid residues of the hNa_V_1.7 channel with an estimated binding affinity of −7.5 kcal/mol, which is adjacent to the transmembrane region (i.e., position: 1696–1721) or membrane segment (i.e., position: 1696–1715 and 1696–1718) of the channel. A predicted interaction could potentially affect CBZ-mediated changes in the magnitude, gating, and Hys_(V)_ of *I*_Na_.

During the last decade, the increased knowledge about causative genetic variants has had a great impact on the development of precision treatments for epilepsy and neuropathic pain. Most epileptogenic Nav variants lie within *SCN1A* (the gene encoding Nav1.1), and a few lie within *SCN2A* (encoding Nav1.2), *SCN3A* (encoding Nav1.3), or *SCN8A* (encoding Nav1.6) [73]. Causative genetic variants in *SCN9A* (encoding Nav1.7), *SCN10A* (encoding Nav1.8) or *SCN11A* (encoding Nav1.9) are linked to neuropathic pain [74]. Among these targets, Neuro-2a cells could express Nav1.2, 1.3, and 1.7 [46]. Our data in Neuro-2a cells revealed that CBZ suppressed the magnitude of *I*_Na(T)_ and *I*_Na(L)_, which may associate with the pathomechanisms of epilepsy susceptibility or neuropathic pain. Additionally, the docking results showed an interaction of the hNaV1.7 channel and the CBZ molecule. We would expect that CBZ might be selected for treating epilepsy with gain-of-function variants in Nav1.2 and 1.3 and neuropathic pain with gain-of-function variants in Nav1.7. However, the lack of direct evidence discriminating the actions of CBZ on specific Nav isoforms is the limitation of this study. It needs to be further investigated.

## 4. Materials and Methods

### 4.1. Chemicals, Drugs, Reagents and Solutions Used in This Work

Carbamazepine (5H-dibenzo[b,f]azepine-5-carboxamide, benzo[b][1]benzazepine-11-carboxamide, C_15_H_12_N_2_O, CBZ, Tegretol^®^), adenosine, adenosine-5-triphosphate (ATP), diazoxide, tefluthrin (Tef), tetraethylammonium chloride (TEA), and tetrodotoxin (TTX) were acquired from Sigma-Aldrich (Merck, Tainan, Taiwan), while ranolazine was from Tocris (Union Biomed; Taipei, Taiwan). To protect CBZ degradation from light illumination [75,76], the stock solution dissolved in ethanol was wrapped in aluminum foil and kept under −20 °C for long-term storage. Unless otherwise stated, culture media (e.g., Dulbecco’s modified eagle’s medium [DMEM]), fetal bovine serum, L-glutamine, and trypsin/EDTA was supplied by HyClone^TM^ (Thermo Fisher; Genechain, Kaohsiung, Taiwan), while other chemical or reagents were of laboratory grade and acquired from standard sources.

The ionic compositions of the extracellular solutions (i.e., HEPES-buffered normal Tyrode’s solution) were as follows (in mM): NaCl 136.5, KCl 5.4, MgCl_2_ 0.53, CaCl_2_ 1.8, glucose 5.5, and HEPES 5.5 (pH 7.4 titrated with NaOH). To record K^+^ currents, the patch pipette was filled up with the internal solution comprising (in mM): K-aspartate 130, KCl 20, KH_2_PO_4_ 1, MgCl_2_ 1, EGTA 0.1, Na_2_ATP 3, Na_2_GTP 0.1, and HEPES 5 (pH 7.2 titrated with KOH). To measure voltage-gated Na^+^ current (*I*_Na_), we replaced K^+^ ions inside the pipette solution with equimolar Cs^+^ ions, and the pH was adjusted to 7.2 by adding CsOH. To record erg-mediated K^+^ current (*I*_K(erg)_), we replaced the bathing solution with a high-K^+^, Ca^2+^-free solution comprised of the following (in mM): KCl 130, NaCl 10, MgCl_2_ 3, glucose 6, and HEPES-KOH buffer 10; pH 7.4. All solutions were prepared by using deionized water which was produced by a Milli-Q^®^ water purification system (Shin Jhih Technology, Tainan, Taiwan).

### 4.2. Cell Preparations

Neuro-2a (N2a), a clonal cell line originally derived from mouse neuroblastoma, was acquired from the Bioresources Collection and Research Center ([BCRC-60026, https://catalog.bcrc.firdi.org.tw/BcrcContent?bid=60026], (accessed on 10 Jan 2022), Hsinchu, Taiwan). This cell line, originally derived from the American Type Culture Collection (ATCC^®^ [CCL-131^TM^]; Manassas, VA, USA), has been established as a model of electrically excitable cells in the studies of electrophysiology and pharmacology [63,64,66,77,78,79]. Cells were maintained in DMEM supplemented with 10% (*v*/*v*) heat-inactivated fetal bovine serum, 2 mM L-glutamine, 1.5 g/L sodium bicarbonate, 0.1 mM non-essential amino acids, and 1.0 mM sodium pyruvate in a humidified atmosphere of CO_2_/air (1:19) at 37 °C [77,80]. The subcultures were made by trypsinization (0.025% trypsin solution [HyClone^TM^] containing 0.01% sodium N,N-diethyldithiocarbamate and EDTA). The measurements were performed five or six days after cells were cultured up to 60–80% confluence.

### 4.3. Patch-Clamp Recordings: Electrophysiological Measurements

During the few hours before the experiments, we gingerly dispersed Neuro-2a cells with 1% trypsin-EDTA solution, and a few drops of cell suspension were thereafter put into a custom-made chamber mounted on the stage of an inverted DM-IL fluorescence microscope (Leica; Major Instruments, Tainan, Taiwan). Cells were kept at room temperature (20–25 °C) in normal Tyrode’s solution, the ionic composition of which was detailed above, and they were settled down to attach the chamber’s bottom. The pipettes that we used to record were prepared from Kimax^®^-51 borosilicate glass tubing with a 1.5–1.8 mm outer diameter (DWK34500-99; Kimble^®^, Sigma-Aldrich, Tainan, Taiwan) using a vertical two-stage puller (PP-83; Narishige, Major Instruments). As they were filled with different internal solution, the electrodes had tip resistances of 3–5 MΩ. During the measurements, the recording pipette was mounted in an air-tight holder which has a suction port on the side, and a silver chloride wire was used to be in contact with the internal solution. We measured ionic currents in the whole-cell configuration of a modified patch–clamp technique with the use of either an Axoclamp-2B (Molecular Devices, Sunnyvale, CA, USA) or an RK-400 amplifier (Bio-Logic, Claix, France), as stated elsewhere [38,42,45,48,67]. The formation of GΩ-seals was commonly made in an all-or-nothing fashion, thereby resulting in a dramatic improvement in signal-to-noise ratio. The liquid junction potentials which occur when the ionic composition in the pipette solution and that of the bath solution are different, were zeroed shorty before GΩ-seal formation was achieved, and the whole-cell data were then corrected. During measurements, we exchanged the solutions through a home-made gravity-driven type of bath perfusion.

The signals were monitored at a given interval and digitally stored online at 10 kHz in an ASUS ExpertBook laptop computer (P2451F; Yuan-Dai, Tainan, Taiwan). For analog-to-digital (A/D) and digital-to-analog (D/A) conversion, a Digidata^®^-1440A equipped with a laptop computer was controlled by pClamp 10.6 runs under Microsoft Windows 7 (Redmond, WA). We low-pass filtered current signals at 2 kHz with an FL-4 four-pole Bessel filter (Dagan, Minneapolis, MN, USA). A variety of pClamp-generated voltage clamp protocols, which include various rectangular or ramp waveforms, were designed and thereafter imposed on the tested cells through D/A conversion in order to evaluate the current–voltage (*I*–*V*) relation or the inactivation curve of ionic currents, as specified in [42]. When pulse–train stimulation was needed, we used an Astro-Med Grass S88X dual output pulse stimulator (Grass, West Warwick, RI, USA).

### 4.4. Data Analyses

To calculate the percentage inhibition of CBZ on the peak or transient component (*I*_Na(T)_) and the sustained or late component (*I*_Na(L)_) of *I*_Na_, Neuro-2a cells were voltage-clamped at −80 mV, a voltage step to −100 mV (30 ms in duration) preceding the abrupt depolarizing pulse (30 ms in duration) to −10 mV was applied to evoked *I*_Na(T)_ and *I*_Na(L)_. The amplitudes of *I*_Na(T)_ or *I*_Na(L)_ achieved at the start or end-pulse of the depolarizing step were thereafter compared after adding different concentrations (3 Μm–1 mM) of CBZ. The CBZ concentration, required to decrease 50% of current amplitude, was determined using a Hill function,
(1)y=(Emax×[CBZ]nH)(IC50nH+[CBZ]nH)
where [*CBZ*] = the CBZ concentration; *IC*_50_ = the concentration required for a 50% decrease; *n_H_* = the Hill coefficient; and *E*_max_ = CBZ-mediated maximal decrease in the amplitude of *I*_Na(T)_ or *I*_Na(L)_.

The *I–V* relationship of *I*_Na(T)_ with or without the application of CBZ was constructed and the data were thereafter fitted with a Boltzmann function given by:(2)IImax={G[1+exp(−(V−Vh)/k)]}×(V−Erev)

In this equation, *V* = membrane potential; *E*_rev_ = the reversal potential of *I*_Na_ (fixed at +45 mV); *G* = *I*_Na_ conductance; *I* = current; and *k* or *V*_h_ = the gating parameters.

### 4.5. Curve-Fitting Approximations and Statistical Analyses

The linear or nonlinear (e.g., exponential or sigmoidal) curve fitting to experimental data sets were performed with the interactive least-squares procedure by different maneuvers, such as Microsoft Excel^®^-embedded “Solver” (Redmond, WA, USA) and OriginPro^®^ 2021 (OriginLab; Scientific Formosa, Kaohsiung, Taiwan). The experimental data are presented as the mean ± standard error of the mean (SEM), with the sizes of independent observations (i.e., cell numbers) from which samples were properly collected. The paired or unpaired *t*-test was made between the two different groups. As the differences among different groups were encountered, we performed either analysis of variance (ANOVA) with or without repeated measures followed by post-hoc Fisher’s least significant difference test. Statistical significance (indicated with * or ** in the figures) was determined at a *p* value of <0.05.

## Figures and Tables

**Figure 1 ijms-23-07892-f001:**
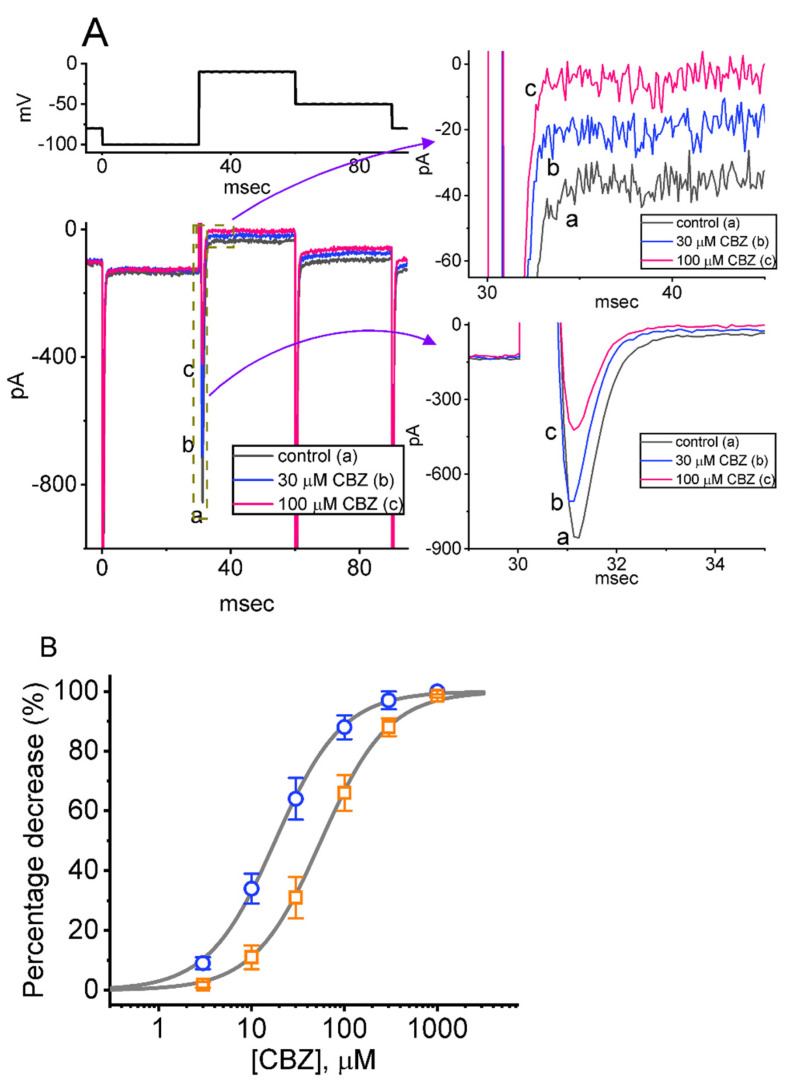
Effect of carbamazepine (CBZ) on voltage-gated Na^+^ current (*I*_Na_) residing in Neuro-2a cells. In this set of measurements, we bathed cells in Ca^2+^-free Tyrode’s solution containing 10 mM tetraethylammonium chloride (TEA) and 0.5 mM CdCl_2_, while the electrode that was used was filled up with an internal solution containing Cs^+^. (**A**) Representative current traces were acquired in the control period (a) (i.e., CBZ was not present) and during cell exposure to 30 μM CBZ (b) or 100 μM CBZ (c). The upper part indicates the voltage clamp protocol that we applied. The graphs demonstrated on the right side of (**A**) denote the expanded records from the left side (broken boxes). (**B**) Concentration–responses curve of CBZ-induced inhibition of peak (transient) *I*_Na_ (*I*_Na(T)_) or sustained (late) *I*_Na_ (*I*_Na(L)_) identified in Neuro-2a cells (mean ± SEM; n = 8–9). The sigmoidal curves drawn represent the goodness-of-fit of the modified Hill equation, as described in Section 4. The IC_50_ values needed for CBZ-mediated inhibition of *I*_Na(T)_ (open orange squares) and *I*_Na(L)_ (open blue circles) were properly estimated to be 56 and 18 μM, respectively. Data analysis was performed by one-way ANOVA (F = 4.7, *p* < 0.05).

**Figure 2 ijms-23-07892-f002:**
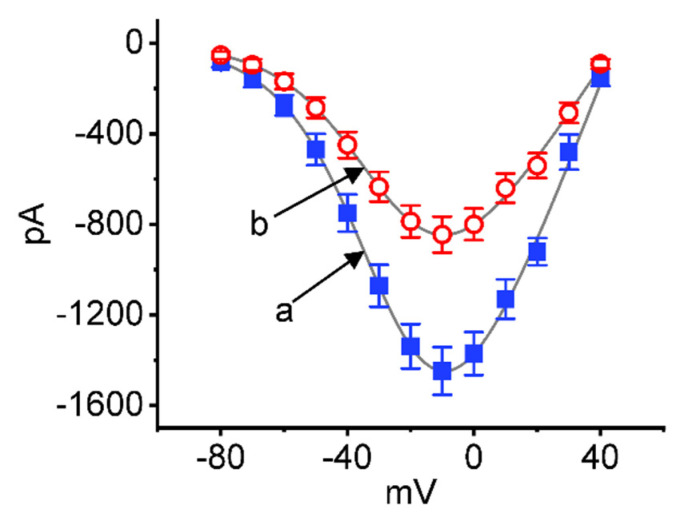
Effect of CBZ on the steady-state current versus voltage (*I–V*) relationship of *I*_Na(T)_ identified in Neuro-2a cells. In these experiments, we voltage-clamped each cell at −80 mV, and various depolarizing command voltages from −80 to +40 mV in 10 mV increments were applied to evoke *I*_Na_. Current amplitude at each depolarizing pulse was taken at the beginning of the voltage pulse. a (Filled blue squares): control (i.e., absence of CBZ); b (open red circles): in the presence of 30 μM CBZ. The smooth gray line over which the data points were overlaid was approximately fitted with a Boltzmann function as elaborated in Section 4.

**Figure 3 ijms-23-07892-f003:**
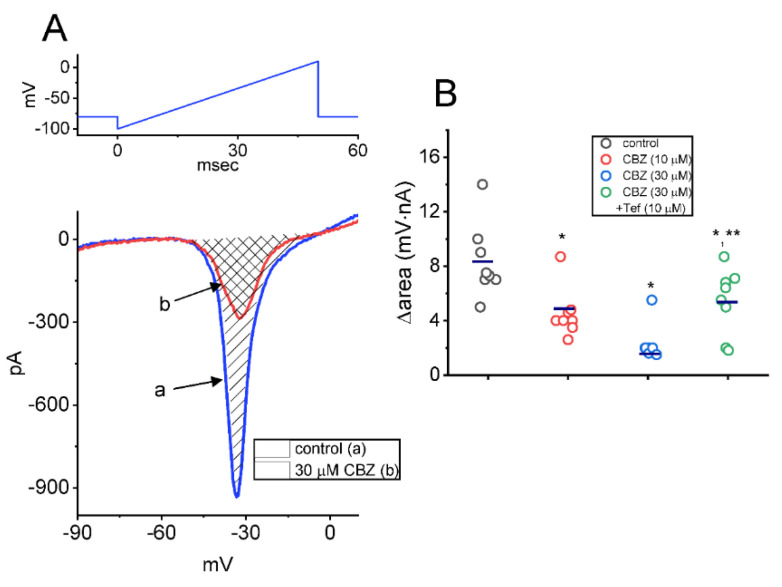
Inhibitory effect of CBZ on window *I*_Na_ (*I*_Na(W)_) evoked by abrupt ascending ramp voltage (V_ramp_) in Neuro-2a cells. This set of whole-cell current recordings was conducted with the examined cell voltage-clamped at −80 mV, and we thereafter imposed the V_ramp_ from −100 to +10 mV for a duration for 50 ms on the cell. (**A**) Representative current traces were acquired in the control period (a, blue color) and during the exposure to 30 μM CBZ (b, red color). The voltage clamp protocol applied is depicted in the upper part, the downward deflection shows the appearance of inward current (i.e., instantaneous *I*_Na(W)_), and the shaded areas indicate the ∆area of *I*_Na(W)_ evoked by short ascending V_ramp_. (**B**) Summary graph demonstrating the effect of CBZ (10 or 30 μM), and CBZ plus tefluthrin (Tef) on the ∆area of *I*_Na(W)_ (mean ± SEM; n = 8 for each point). Each horizontal bar indicates the mean value. The ∆area (i.e., shaded region in (**A**)) was calculated at the voltage ranging between −50 and +10 mV during the upsloping V_ramp_. The analysis was made by one way ANOVA (F value = 4.8; *p* < 0.05). * Significantly different from control (t value = 3.2; *p* < 0.05) and ** significantly different from CBZ (30 μM) alone group (t value = 3.3; *p* < 0.05).

**Figure 4 ijms-23-07892-f004:**
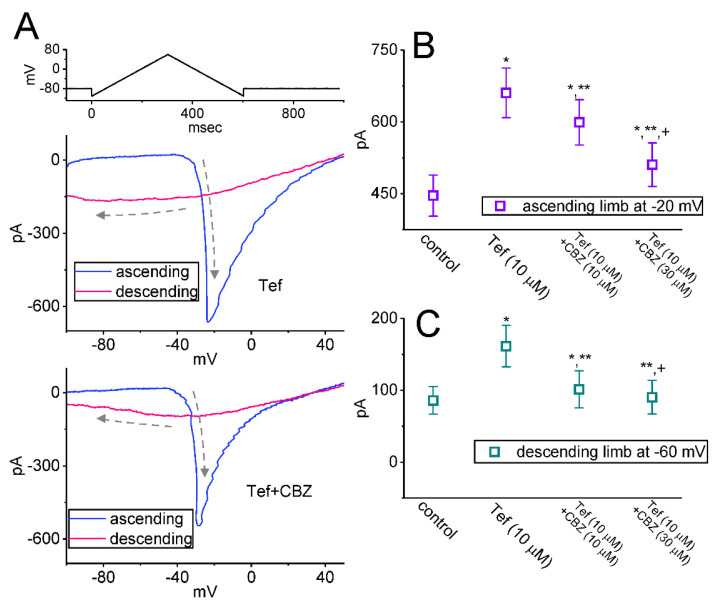
Modification by CBZ and CBZ plus tefluthrin (Tef) on persistent *I*_Na_ (*I*_Na(P)_) activated in response to upright isosceles-triangular ramp voltage (V_ramp_). This set of recordings was applied to mimic the depolarizing or repolarizing slopes of bursting patterns in excitable cells. (**A**) Representative current traces were activated by isosceles-triangular V_ramp_ for a duration of 600 ms or with a ramp speed of ±0.5 mV/ms (indicated in the uppermost part). The blue color in the upper or lower part of (**A**) shows the current trace activated by the ascending limb of V_ramp_ acquired in the presence of 10 μM tefluthrin (Tef) or 10 μM Tef plus 30 μM CBZ, respectively, whereas the red color is the trace evoked by the V_ramp_’s descending limb. The dashed curves indicate the direction of the current over which time goes during the activation of the triangular V_ramp_. Notably, there is a voltage-dependent hysteresis (Hys_(V)_) (i.e., figure of eight [∞] configuration) of *I*_Na(P)_ activated by the isosceles-triangular V_ramp_ during cell exposure to Tef (10 μM) or Tef (10 μM) plus CBZ (30 μM). In (**B**,**C**), summary graphs, respectively, demonstrate the inhibitory effect of Tef (10 μM) and Tef (10 μM) plus CBZ (10 or 30 μM) on the amplitude of *I*_Na(P)_ activated by the upsloping (at −20 mV) and downsloping (at −60 mV) limbs of the triangular V_ramp_ (mean ± SEM; n = 8 for each point). The horizontal bars in (**B**,**C**) indicate the mean values. The analysis was made by one-way ANOVA (F value = 4.9; *p* < 0.05). * Significantly different from controls (*p* < 0.05), ** significantly different from Tef (10 μM) alone group (t value = 3.6; *p* < 0.05), and ^+^ significantly different from Tef (10 μM) plus CBZ (10 μM) group (t value = 3.5; *p* < 0.05).

**Figure 5 ijms-23-07892-f005:**
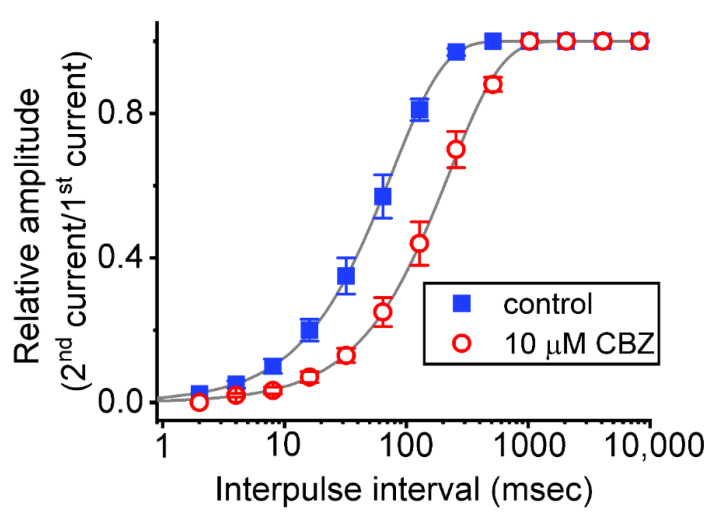
Modification by CBZ on the recovery of *I*_Na(T)_ inactivation evoked by varying interpulse intervals with a geometrics-based progression. In these measurements, we put cells bathed in Ca^2+^-free Tyrode’s solution, while we filled up the recording pipette with Cs^+^-enriched solution. Each tested cell was depolarized from −80 to −10 mV for a duration of 30 ms, and subsequently different interpulse durations with a geometrics-based progression (common ratio = 2) were applied to it. The relative amplitude of peak *I*_Na_ was measured as a ratio of the second peak amplitude divided by the first peak amplitude. The recovery time course (indicated by the smooth gray line) in the presence and presence of 10 μM CBZ was noticed to display an exponential increase as a function of the interpulse interval, with a time constant of 75.3 and 223 ms, respectively. Notably, the horizontal axis is illustrated with a logarithmic scale. Each point represents the mean ± SEM (n = 8). The analysis was made by one way ANOVA (F value = 5.3, *p* < 0.05).

**Figure 6 ijms-23-07892-f006:**
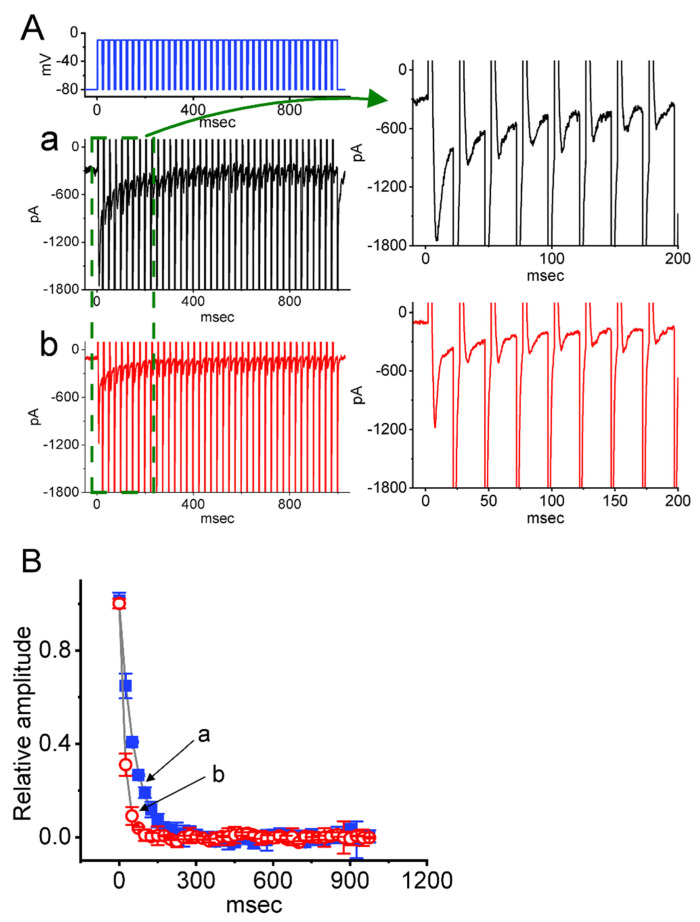
Effect of CBZ on *I*_Na(T)_ evoked by a train of depolarizing pulses in Neuro-2a cells. The train applied consists of 40–20 ms pulses (stepped to −10 mV) separated by 5 ms intervals at −80 mV for a duration of 1 s. (**A**) Representative current traces were acquired in the control period (**a**, CBZ was not present, black color) and during cell exposure to 30 μM CBZ (**b**, red color). The voltage clamp protocol is illustrated in the uppermost part. To provide a single *I*_Na_ trace, the right side of (**A**) shows the expanded records from a broken green box on the left side. (**B**) The relationship of *I*_Na(T)_ versus the pulse train duration in the absence (a, filled blue squares) and presence (b, open red circles) of 30 μM CBZ (mean ± SEM; n = 7 for each point). The continuous smooth gray lines over which the data points are overlaid are fitted by a single exponential. Notably, cell exposure to CBZ can enhance the time course of *I*_Na(T)_ inactivation activated in response to a train of depolarizing command voltages. The analysis was made by one way ANOVA (F value = 5.9, *p* < 0.05).

**Figure 7 ijms-23-07892-f007:**
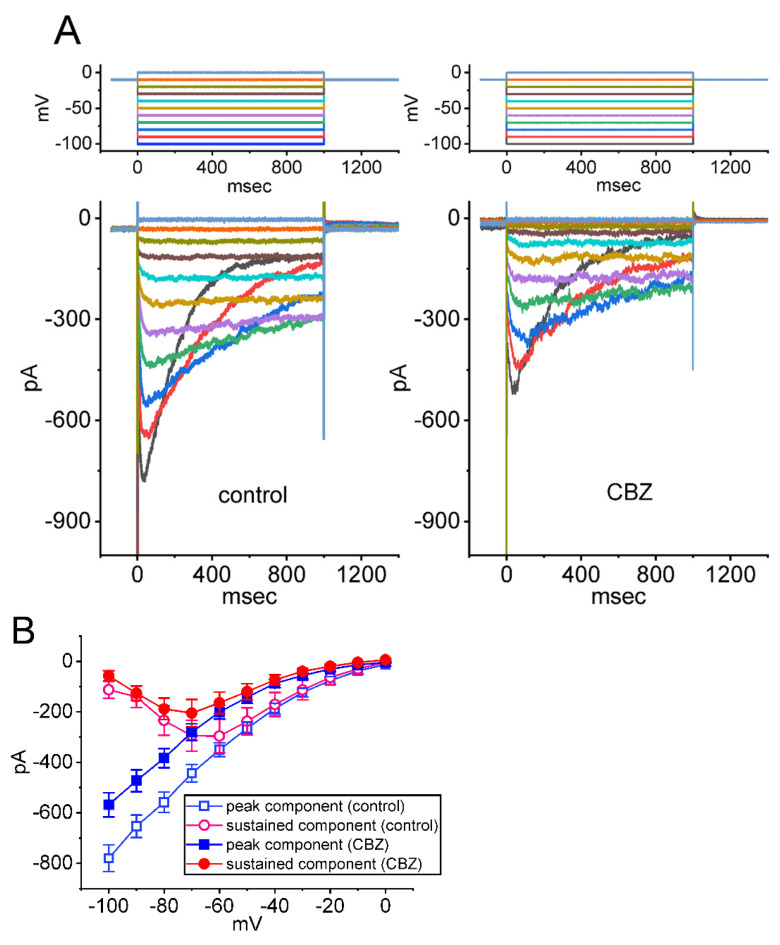
Inhibitory effect of CBZ on the steady-state *I-V* relationship of hyperpolarization-activated *erg*-mediated K^+^ current (*I*_K(erg)_) identified in Neuro-2a cells. In these experiments, we put cells in a high-K^+^, Ca^2+^-free solution which contained 1 μM TTX and 0.5 mM CdCl_2_, and we filled up the recording pipette with a K^+^-enriched solution. The tested cell was voltage-clamped at −10 mV and a series of command potentials ranging between −100 and 0 mV for 1 s was imposed on it. (**A**) Representative current traces were acquired in the control period (left side, absence of CBZ) and during exposure to 100 μM CBZ (right side). The voltage clamp protocol is illustrated in the upper parts of current traces with or without the application of CBZ. (**B**) Average *I–V* relationships of the peak (blue squares) or sustained component (red circles) of *I*_K(erg)_ acquired in the control period (open symbols) and during exposure to 100 μM CBZ (filled symbols). The peak and sustained amplitudes of deactivating *I*_K(erg)_ were obtained at the beginning and end-pulse of each step command applied. Each data point indicates the mean ± SEM (n = 8). The statistical analyses made with or without the presence of CBZ among different levels of membrane potentials given were performed with two-way ANOVA analysis (F value = 6.3; *p* < 0.05).

**Figure 8 ijms-23-07892-f008:**
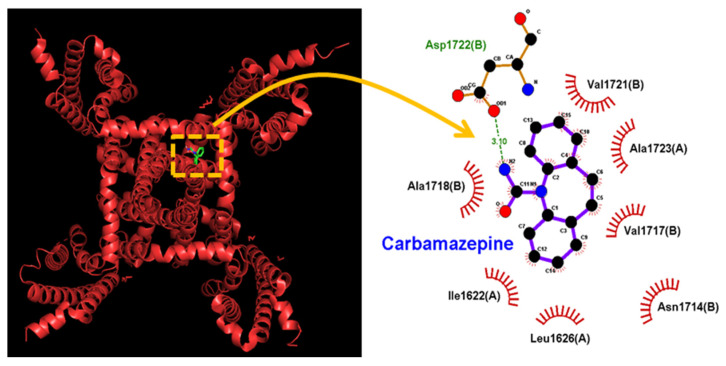
Docking results of hNa_V_1.7 results and carbamazepine (CBZ). The protein structure of the hNa_V_1.7 channel was obtained from PDB (PDB ID: 5EK0), while the chemical structure of CBZ was acquired from PubChem (Compound CID: 2554). The structure of the hNa_V_1.7 channel was docked by the CBZ molecule through PyRx (http://pyrx.sourceforge.io/ accessed on 12 July 2022). The diagram of interaction between the hNa_V_ channel and the CBZ molecule (at the right side) was generated by LigPlot^+^ (http://www.ebi.ac.uk/thornton-srv/software/LIGPLOT/ accessed on 12 July 2022). Of note, the red arcs with spokes that radiate toward the ligand (i.e., carbamazepine, CBZ) indicate hydrophobic contact, while the green dashed line is the formation of hydrogen bond with a distance of 3.10 Å.

## Data Availability

The original data are available upon reasonable request to the corresponding author.

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
