# Peer review of "Characterization in Inhibitory Effectiveness of Carbamazepine in Voltage-Gated Na+ and Erg-Mediated K+ Currents in a Mouse Neural Crest-Derived (Neuro-2a) Cell Line"

_ijms, 2022, doi:10.3390/ijms23147892_

Round 1
Reviewer 1 Report
This manuscript highlights the effects of Carbamazepine in inhibiting sodium channels and Kerg channels. Carbamazepine is known to be an anticonvulsant drug and also used in the treatment of neuropathic pain. Recording the effects of Carbamazepine in Neura2a cell line explains the role on different ion channels. The experiments are very sound and the results are clearly presented.
Author Response
Reviewer 1:
This manuscript highlights the effects of Carbamazepine in inhibiting sodium channels and Kerg channels. Carbamazepine is known to be an anticonvulsant drug and also used in the treatment of neuropathic pain. Recording the effects of Carbamazepine in Neura2a cell line explains the role on different ion channels. The experiments are very sound and the results are clearly presented.
Ans: Thank you very much for your affirmation.
Reviewer 2 Report
Manuscript by Wu et al “Characterization of inhibitory effectiveness of carbamazepine in voltage gated Na+ currents and erg-mediated K+ currents in a mouse neural crest (Neuro-2a) cell line
Major comments:
The study aims to characterize the effectiveness and specificity of carbamazepine on several components of sodium current carried through voltage gated sodium channels, presumably to more accurately determine the actions of this drug used as an anticonvulsant in the treatment of epilepsy syndromes and neuropathic pain.
The authors employ a neural crest cell line (Neuro-2a) to accomplish this. There is no mention of the reason why this cell line is used to study the actions of a drug used to treat conditions for which neurons in the brain or peripheral nerve are affected. To that end, what sodium channels are expressed in this cell line? Are these sodium channels those targeted by mutations in brain sodium channels, or peripheral nerve sodium channels? A study in which total sodium current is the target of investigation with no means to discriminate the actions of the drug on specific isoforms severely limits the relevance of the experimental findings to the stated aim of increasing the understanding of the specific actions of carbamazepine on mutations in sodium channels that cause epilepsy syndromes or neuropathic pain.
In the discussion it is mentioned that two previous studies suggest that sodium channel transcripts in this cell line are dominated by SCN3A (NaV1.3) or SCN9A (NaV1.7), and that these reports are thus in conflict with one another. Uncertainty of the isoforms expressed in this cell line is a significant limitation towards an interpretation of the effects of carbamazepine on voltage-gated sodium channel defects associated with epilepsy or neuropathic pain. Additionally, the authors fail to point out the fact that SCN3A/NaV1.3 is not a primary target for epilepsy mutations, or that SCN9A/NaV1.7 is a primary target in neuropathic pain in their discussion, but leave this as open question as to the ionic mechanism of carbamazepine action in this cell line. Again, relevance is limited.
A related concern is the emphasis on the effect of carbamazepine on the late, or persistent current in sodium channels. This effect is documented for the Neuro-2a cell line. Do epilepsy mutations or those associated with neuropathic pain cause gating defects that target the late, persistent current? The authors do not address the findings of investigation using heterologous expression of sodium channels for which mutations have been identified in epileptic syndromes or neuropathic pain at all, including whether or not those mutations cause gating defects that influence the late, persistent sodium current. In other words, while the target sodium channel is not known for carbamazepine on Neuro-2a sodium current, is the effect relevant to either epilepsy syndrome or neuropathic pain based on what is known about gating defects in these syndromes? This is not addressed.
For erg-mediated potassium currents, a similar problem regarding relevance is observed in this manuscript. Why are these currents being studied? There is no mention of their involvement in epilepsy syndromes, neuropathic pain, or any other condition by the authors in this manuscript.
Side effects of carbamazepine are noted. How a study of the impact of this drug on transient, late, persistent or window current might influence these side effects is not addressed. There is mention of a possible relationship between the effect of carbamazepine on late sodium current (decreased) and the side effect of carbamazepine to cause lowered extracellular sodium. The authors do acknowledge that the experimental evidence in this manuscript is not sufficient to provide causality at this point. No other side effects are mentioned with respect to the findings of this study.
Taken together these concerns point out that this investigation is one in which an initial characterization of the ionic currents modifiable by an anticonvulsant, carbamazepine, is the goal. There is no support for describing these results in terms of their relevance to channelopathies or other health conditions, at this point.
Minor comments:
Page 2 - The use of 0.5 mM CdCl2 in the recording bath is of concern. Cadmium is capable of blocking the central pore at mM concentrations, and can form metal bridges with the sulfhydryl groups of cysteine residues. Its effects are not limited to chelation of calcium and why other agents such as EGTA were not compared is not clear.
Page 3 – The authors state that the transient sodium current was sensitive to TTX at 1 µM, but give no data to support this claim, or quantify this sensitivity.
Page 4 – The I/V relations for Neuro-2a sodium current in the absence of presence of carbamazepine are shown. The interpretation is that steady state activation is not affected, only current amplitude. Activation is measured as probability and should be presented from a calculation using I/V relations to determine conductance (g/V) relations as with a straightforward Boltzmann equation calculation. That equation is given later in the Methods. But is not used in the results.
Page 9 – What is the basis for the conclusion that the potassium currents under investigation are IKerg (only)? The interpretation that carbamazepine could inhibit erg currents in response to long-lasting hyperpolarization is stated (later) in discussion with respect to cardiac arrythmia, as a phenomenon worth further study (again there is a limited attempt by the authors to provide a relevance of the experimental findings to specific health conditions).
Page 12-13, modeling data and discussion. The pdb used is for a prokaryotic Ms channel without any substantive reason except that given by the authors of the paper describing its structure. There are numerous structures from cryo-EM for eukaryotic sodium channels for which at least inactivation mechanisms (and their underlying structural basis) might be more relevant to the aim of increasing our understanding of the way carbamazepine interacts with eukaryotic sodium channels. Not surprisingly, then, there is a description of docking in which carbamazepine interacts with some amino acids, but no further interpretation of that result based on any known mechanisms of channel activity for which these interactions might play a role.
Reviewer 3 Report
The manuscript “Characterization in inhibitory effectiveness of carbamazepine 2 in voltage-gated Na+ and erg-mediated K+ currents in a mouse 3 neural crest-derived (Neuro-2a) cell line” by Wu et al is a research article which describes the effects of carbamazepine on voltage-gated Na+ and erg-mediated K+ currents in Neuro-2a cells. Generally, the study is important in this filed and scientifically sound and contains essential findings. This topic is also of importance for treatment of pathological pain. The manuscript has been well organized and written. However, I have some concerns on the paper.
The statistical significance was assessed using Student’s t-test or one/two-way ANOVA.. The values of “F” and “t” should be given.
It would be better to show all data plots in bar graph (Figure 3 and 4), if possible. All of them could provide much more information.
In Figure 7, it would be better to combine the plot for control and that of CBZ in one graph. Then, the readers can easily compare the amplitude of erg-mediated K+ currents.
